# Deep-Learning-Based Pupil Center Detection and Tracking Technology for Visible-Light Wearable Gaze Tracking Devices

**Wei-Liang Ou, Tzu-Ling Kuo, Chin-Chieh Chang and Chih-Peng Fan ***

Department of Electrical Engineering, National Chung Hsing University, 145 Xingda Rd., South Dist., Taichung City 402, Taiwan; soft96123@gmail.com (W.-L.O.); claire5883claire5883@gmail.com (T.-L.K.); g107064065@mail.nchu.edu.tw (C.-C.C.)
* Correspondence: cpfan@dragon.nchu.edu.tw

**Abstract:** In this study, for the application of visible-light wearable eye trackers, a pupil tracking methodology based on deep-learning technology is developed. By applying deep-learning object detection technology based on the You Only Look Once (YOLO) model, the proposed pupil tracking method can effectively estimate and predict the center of the pupil in the visible-light mode. By using the developed YOLOv3-tiny-based model to test the pupil tracking performance, the detection accuracy is as high as 80%, and the recall rate is close to 83%. In addition, the average visible-light pupil tracking errors of the proposed YOLO-based deep-learning design are smaller than 2 pixels for the training mode and 5 pixels for the cross-person test, which are much smaller than those of the previous ellipse fitting design without using deep-learning technology under the same visible-light conditions. After the combination of calibration process, the average gaze tracking errors by the proposed YOLOv3-tiny-based pupil tracking models are smaller than 2.9 and 3.5 degrees at the training and testing modes, respectively, and the proposed visible-light wearable gaze tracking system performs up to 20 frames per second (FPS) on the GPU-based software embedded platform.

**Keywords:** deep-learning; YOLOv3-tiny; visible-light; pupil tracking; gaze tracker; wearable eye tracker

## 1. Introduction

Recently, wearable gaze tracking devices (or called eye trackers) have started to become more widely used in human-computer interaction (HCI) applications. To evaluate people's attention deficit, vision, and cognitive information and processes, gaze tracking devices have been well-utilized in this field [1–6]. In [4], the eye-tracking technology was a vastly applied methodology to examine hidden cognitive processes. The authors developed a program source code debugging procedure that was inspected by using the eye-tracking method. The eye-tracking-based analysis was elaborated to test the debug procedure and record the major parameters of eye movement. In [5], the eye-movement tracking device was used to observe and analyze a complex cognitive process for C# programming. By evaluating the knowledge level and eye movement parameters, the test subjects were involved to analyze the readability and intelligibility of the query and method at the Language-Integrated Query (LINQ) declarative query of the C# programming language. In [6], the forms and efficiency of debugging parts of software development were observed by tracking eye movements with the participation of test subjects. The routes of gaze for the test subjects were observed continuously during the procedure, and the coherences were decided by the assessment of the recorded metrics.

Many up-to-date consumer products have applied embedded gaze tracking technology in their designs. Through the use of gaze tracking devices, the technology improves the functions and applications of intuitive human-computer interaction. Gaze tracking designs use an image sensor based on near-infrared (NIR), which was utilized in [7–16]. Some previous eye-tracking designs used high-contrast eye images, which are recorded by

active NIR image sensors. NIR image sensor-based technology can achieve high accuracy when used indoors. However, the NIR-based design is not suitable for use outdoors or with sunlight. Moreover, long-term operation may have the potential to hurt the user's eyes. When eye trackers are considered for general and long-term use, visible-light pupil tracking technology becomes a suitable solution.

Due to the characteristics of the selected sensor module, the contrast of the visible-light eye image may be insufficient, and the nearfield eye image may contain some random types of noise. In general, visible-light eye images usually include insufficient contrast and large random noise. In order to overcome this inconvenience, some other methods [17–19] have been developed for visible-light-based gaze tracking design. For the previous wearable eye tracking design in [15,19] with the visible-light mode, the estimation of pupil position was easily disturbed by light and shadow occlusion, and the prediction error of the pupil position, thus, becomes serious. Figure 1 illustrates some of the visible-light pupil tracking results by using the previous ellipse fitting-based methods in [15,19]. Thus, a large pupil tracking error will lead to inaccuracy of the gaze tracking operation. In [20,21], for indoor applications, the designers developed eye tracking devices based on visible-light image sensors. A previous study [20] performed experiments indoors, but showed that the device could not work normally outdoors. When the wearable eye-tracking device is correctly used in all indoor and outdoor environments, the method based on the visible-light image sensor is sufficient.

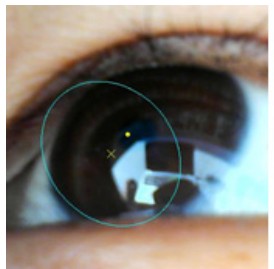 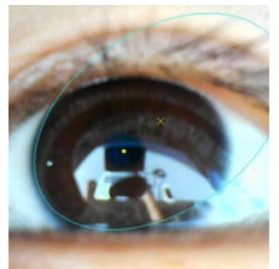 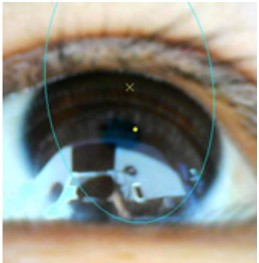

**Figure 1.** Visible-light pupil tracking results by the previous ellipse fitting-based method.

*Related Works*

In recent years, the progress of deep-learning technologies has become very important, especially in the field of computer vision and pattern recognition. Through deep-learning technology, inference models can be trained to realize real-time object detection and recognition. For the deep-learning-based designs in [22–37], the eye tracking systems could detect the location of eyes or pupils, regardless of using the visible-light or near-infrared image sensors. Table 1 describes the overview of the deep-learning-based gaze tracking designs. From the viewpoints of device setup, the deep-learning-based gaze tracking designs in [22–37] can be divided into two different styles, which include the non-wearable and the wearable types.

For the deep-learning-based non-wearable eye tracking designs, in [22], the authors proposed a deep-learning-based gaze estimation scheme to estimate the gaze direction from a single face image. The proposed gaze estimation design was based on applying multiple convolutional neural networks (CNN) to learn the model for gaze estimation from the eye images. The proposed design provided accurate gaze estimation for users with different head poses by including the head pose information into the gaze estimation framework. Compared with the previous methods, the developed gaze estimation design increased the accuracy of appearance-based gaze estimation under head pose variations.

In [23], the purpose of this work was to use CNNs to classify eye tracking data. Firstly, a CNN was used to classify two different web interfaces for browsing news data. Secondly, a CNN was utilized to classify the nationalities of users. In addition, the data-preprocessing and feature-engineering techniques were applied. In [24], for emerging consumer electronic products, an accurate and efficient eye gaze estimation design was important, and

the products were required to remain reliable under difficult environments with low-power consumption and low hardware cost. In this paper, a new hardware-friendly CNN model with minimal computational requirements was developed and assessed for efficient appearance-based gaze estimation. The proposed CNN model was tested and compared against existing appearance-based CNN approaches, achieving better eye gaze accuracy with large reductions of computational complexities.

In [25], iRiter technology was developed, which can assist paralyzed people to write on screen by using only their eye movements. iRiter detects precisely the movement of the eye pupil by referring to the reflections of external near-infrared (IR) signals. Then, the deep multilayer perceptron model was used for the calibration process. By the position vector of pupil area, the coordinates of the pupil were mapped to the given gaze position on the screen.

In [26], the authors proposed a gaze zone estimation design by using deep-learning technology. Compared with traditional designs, the developed method did not need the procedure of calibration. In the proposed method, a Kinect was used to capture the video of a computer user, a Haar cascade classifier was applied to detect the face and eye regions, and data on the eye region was used to estimate the gaze zones on the monitor by the trained CNN. The proposed calibration-free method performed a high accuracy to be applied for human–computer interaction.

In [27], which focused on healthcare applications, the high accuracy in semantic segmentation of medical images were important; a key issue for training the CNNs was obtaining large-scale and precisely annotated images, and the authors sought to address the lack of annotated data used with eye tracking method. The hypothesis was that segmentation masks generated to help eye tracking would be very similar to those rendered by hand annotation, and the results demonstrated that the eye tracking method created segmentation masks, which were suitable for deep-learning-based semantic segmentation.

In [29], for the conditions of unconstrained gaze estimation, most of the gaze datasets were collected under laboratory conditions, and the previous methods were not evaluated across multiple datasets. In this work, the authors studied key challenges, including target gaze range, illumination conditions, and facial appearance variation. The study showed that image resolution and the use of both eyes affected gaze estimation performance. Moreover, the authors proposed the first deep appearance-based gaze estimation method, i.e., GazeNet, to improve the state of the art by 22% for the cross-dataset evaluation.

In [30], a subconscious response influenced the estimation of human gaze from factors related to the human mental activity. The previous methodologies only based on the use of gaze statistical modeling and classifiers for assessing images without referring to a user's interests. In this work, the authors introduced a large-scale annotated gaze dataset, which was suitable for training deep-learning models. Then, a novel deep-learning-based method was used to capture gaze patterns for assessing image objects with respect to the user's preferences. The experiments demonstrated that the proposed method performed effectively by considering key factors related to the human gaze behavior.

In [31], the authors proposed a two-step training network, which was named the Gaze Estimator, to raise the gaze estimation accuracy on mobile devices. At the first step, an eye landmarks localization network was trained on the 300W-LP dataset, and the second step was to train a gaze estimation model on the GazeCapture dataset. In [28], the network localized the eye precisely on the image and the CNN network was robust when facial expressions and occlusion happened; the inputs of the gaze estimation network were eye images and eye grids.

In [32], since many previous methods were based on a single camera, and focused on either the gaze point estimation or gaze direction estimation, the authors proposed an effective multitask method for the gaze point estimation by multi-view cameras. Specifically, the authors analyzed the close relationship between the gaze point estimation and gaze direction estimation, and used a partially shared convolutional neural networks to estimate

the gaze direction and gaze point. Moreover, the authors introduced a new multi-view gaze tracking data set consisting of multi-view eye images of different subjects.

In [33], the U2Eyes dataset was introduced, and U2Eyes was a binocular database of synthesized images regenerating real gaze tracking scripts with 2D/3D labels. The U2Eyes dataset could be used in machine and deep-learning technologies for eye tracker designs. In [34], for using appearance-based gaze estimation techniques, only using a single camera for the gaze estimation will limit the application field to short distance. To overcome this issue, the authors developed a new long-distance gaze estimation design. In the training phase, the Learning-based Single Camera eye tracker (LSC eye tracker) acquired gaze data by a commercial eye tracker, the face appearance images were captured by a long-distance camera, and deep convolutional neural network models were used to learn the mapping from appearance images to gazes. In the application phase, the LSC eye tracker predicted gazes based on the acquired appearance images by the single camera and the trained CNN models with effective accuracy and performance.

In [35], the authors introduced an effective eye-tracking approach by using a deep-learning-based method. The location of the face relative to the computer was obtained by detecting color from the infrared LED with OpenCV, and the user's gaze position was inferred by the YOLOv3 model. In [36], the authors developed a low-cost and accurate remote eye-tracking device which used an industrial prototype smartphone with integrated infrared illumination and camera. The proposed design used a 3D gaze-estimation model which enabled accurate point-of-gaze (PoG) estimation with free head and device motion. To accurately determine the input eye features, the design applied the convolutional neural networks with a novel center-of-mass output layer. The hybrid method, which used artificial illumination, a 3D gaze-estimation model, and a CNN feature extractor, achieved better accuracy than the existing eye-tracking methods on smartphones.

On the other hand, for the deep-learning-based wearable eye tracking designs, in [28], the eye tracking method with video-oculography (VOG) images needed to provide an accurate localization of the pupil with artifacts and under naturalistic low-light conditions. The authors proposed a fully convolutional neural network (FCNN) for the segmentation of the pupil area that was trained on 3946 hand-annotated VOG images. The FCNN output simultaneously performed pupil center localization, elliptical contour estimation, and blink detection with a single network on commercial workstations with GPU acceleration. Compared with existing methods, the developed FCNN-based pupil segmentation design was accurate and robust for new VOG datasets. In [34], to conquer the obstructions and noises in nearfield eye images, the nearfield visible-light eye tracker design utilized the deep-learning-based gaze tracking technologies to solve the problem.

In order to make the pupil tracking estimation more accurate, this work uses the YOLOv3-tiny [38] network-based deep-learning design to detect and track the position of the pupil. The proposed pupil detector is a well-trained deep-learning network model. Compared with the previous designs, the proposed method reduces the pupil tracking errors caused by light reflection and shadow interference in visible-light mode, and will also improve the accuracy of the calibration process of the entire eye tracking system. Therefore, the gaze tracking function of the wearable eye tracker will become powerful under visible-light conditions.

**Table 1.** Overview of the deep-learning-based gaze tracking designs.

| Deep-learning-based Methods | Operational Mode/Setup | Dataset Used | Eye/Pupil Tracking Method | Gaze Estimation and Calibration Scheme |
|---|---|---|---|---|
| Sun et al. [22] | Visible-light mode /Non-Wearable | UT Multiview | Uses a facial landmarks-based design to locate eye regions | Multiple pose-based and VGG-like CNN models for gaze estimation |
| Lemley et al. [24] | Visible-light mode /Non-Wearable | MPII Gaze | The CNN-based eye detection | Joint eye-gaze CNN architecture with both eyes for gaze estimation |
| Cha et al. [26] | Visible-light mode /Non-Wearable | Self-made dataset | Uses the OpenCV method to extract the face region and eye region | GoogleNetV1 based gaze estimation |
| Zhang et al. [29] | Visible-light mode /Non-Wearable | MPII Gaze | Uses face alignment and 3D face model fitting to find eye zones | VGGNet-16-based GazeNet for gaze estimation |
| Lian et al. [32] | Visible-light mode /Non-Wearable | ShanghaiTechGaze and MPII Gaze | Uses the ResNet-34 model for eye features extraction | CNN-based multi-view and multi-task gaze estimation |
| Li et al. [34] | Visible-light mode /Non-Wearable | Self-made dataset | Uses the OpenFace CLM-framework/Face++/YOLOv3 to extract facial ROI and landmarks | Infrared-LED based calibration |
| Rakhmatulin et al. [35] | Visible-light mode /Non-Wearable | Self-made dataset | Uses YOLOv3 to detect the eyeball and eye's corners | Infrared-LED based calibration |
| Brousseau et al. [36] | Infrared mode /Non-Wearable | Self-made dataset | Eye-Feature locator CNN model | 3D gaze-estimation model-based design |
| Yiu et al. [28] | Near-eye infrared mode /Wearable | German center for vertigo and balance disorders | Uses a fully convolutional neural network for pupil segmentation and uses ellipse fitting to locate the eye center | 3D eyeball model, marker, and projector-assisted-based calibration |
| Proposed design | Near-eye visible-light mode/Wearable | Self-made dataset | Uses the YOLOv3-tiny-based lite model to detect the pupil zone | Marker and affine transform-based calibration |

The main contribution of this study is that the pupil location in nearfield visible-light eye images is detected precisely by the YOLOv3-tiny-based deep-learning technology. To our best knowledge from surveying related studies, the novelty of this work is to provide the leading reference design of deep-learning-based pupil tracking technology using the nearfield visible-light eye images for the application of wearable gaze tracking devices. Moreover, the other novelties and advantages of the proposed deep-learning-based pupil tracking technology are described as follows:

(1) By using a nearfield visible-light eye image dataset for training, the used deep-learning models achieved real-time and accurate detection of the pupil position at the visible-light mode.

(2) The proposed design detects the position of the pupil's object at any eyeball movement condition, which is more effective than the traditional image processing methods without deep-learning technologies at the near-eye visible-light mode.

(3) The proposed pupil tracking technology can overcome efficiently the light and shadow interferences at the near-eye visible-light mode, and the detection accuracy of a pupil's

location is higher than that of previous wearable gaze tracking designs without using deep-learning technologies.

To enhance detection accuracy, we used the YOLOv3-tiny-based deep-learning model to detect the pupil's object in the visible-light near-eye images. We also compared its detection results to the other designs, which include the methods without using deep-learning technologies. The pupil detection performance was evaluated in terms of precision, recall, and pupil tracking errors. By the proposed YOLOv3-tiny-based deep-learning-based method, the trained YOLO-based deep-learning models achieve enough precision with small average pupil tracking errors, which are less than 2 pixels for the training mode and 5 pixels for the cross-person test under the near-eye visible-light conditions.

## 2. Background of Wearable Eye Tracker Design

Figure 2 depicts the general two-stage operational flow of the wearable gaze tracker with the four-point calibration scheme. The wearable gaze tracking device can use the NIR or visible-light image sensor-based gaze tracking design, which can apply appearance-, model-, or learning-based technology. First, the gaze tracking device estimates and tracks the pupil's center to prepare the calibration process. For the NIR image sensor-based design, for example, the first-stage computations, which involve the processes for pupil location, extraction of the pupil's region of interest (ROI), the binarization process with the two-stage scheme in pupil's ROI, and the fast scheme of ellipse fitting with binarized pupil's edge points, are utilized [15]. Then, the pupil's center can be detected effectively. For the visible-light image sensor-based design, for example, by using the gradients-based iris center location technology [21] or the deep-learning-based methodology [37], the gaze tracking device realizes an effective iris center tracking capability. Next, at the second processing stage, the function and process for calibration and gaze prediction can be done. To increase the accuracy of gaze tracking, the head movement effect can be compensated by the motion measurement device [39]. In the following section, for the proposed wearable gaze tracking design, the first-stage processing procedure uses the deep-learning-based pupil tracking technology by using a visible-light nearfield eye image sensor. Next, the second-stage process achieves the function of calibration for the gaze tracking and prediction work.

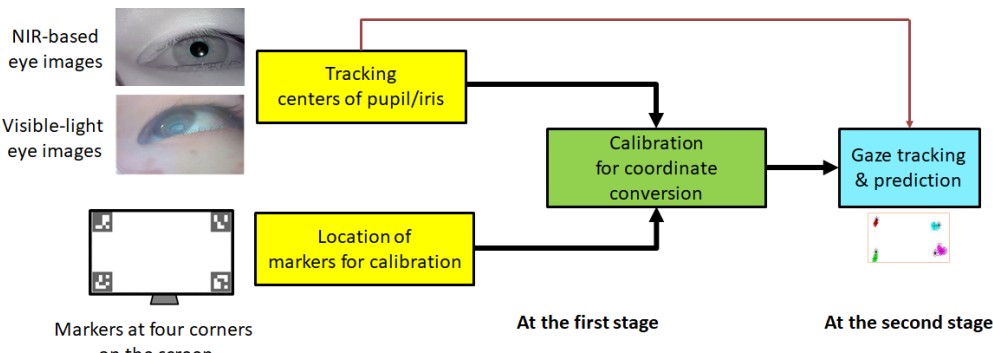

**Figure 2.** The general processing flow of the wearable gaze tracker.

## 3. Proposed Methods

The proposed wearable eye tracking method is divided into several steps as follows: (1) Collecting the visible-light nearfield eye images in datasets and labeling these eye image datasets; (2) Choosing and designing the deep-learning network architecture for pupil object detection; (3) Training and testing the trained inference model; (4) Using the deep-learning model to infer and detect the pupil's object, and estimate the center coordinate of pupil box; (5) Following the calibration process and predicting the gaze points. Figure 3 depicts the computational process of the proposed visible-light gaze tracking design based on the YOLO deep-learning model. In the proposed design, a visible-light camera module is used to capture nearfield eye images. The design uses a deep-learning network

based on YOLO to train an inference model to detect the object of pupil, and then, the proposed system calculates the pupil center from the detected pupil box for the subsequent calibration and gaze tracking process.

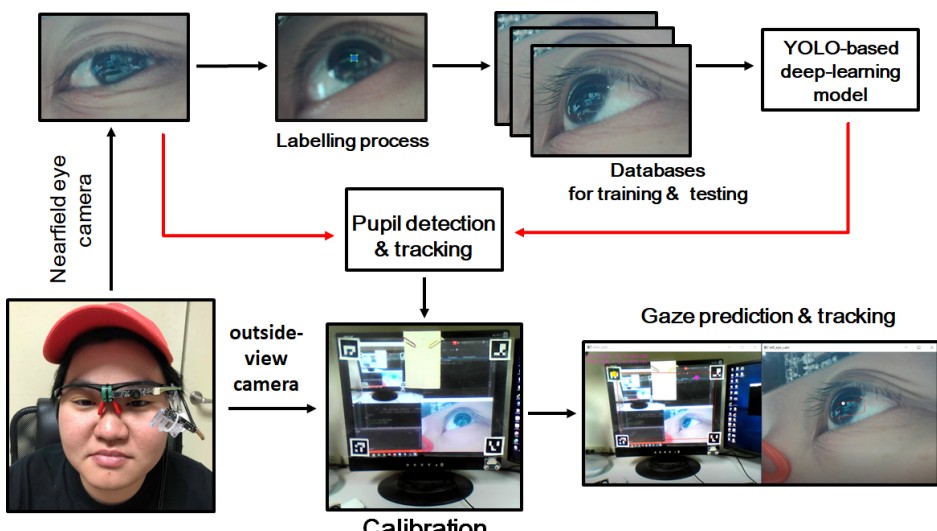

**Figure 3.** The operational flow of the proposed visible-light based wearable gaze tracker.

Figure 4 demonstrates the scheme of the self-made wearable eye tracking device. In the proposed wearable device, both the outward-view scene camera and the nearfield eye camera are required to provide the whole function of the wearable gaze tracker. The proposed design uses deep-learning-based technology to detect and track the pupil zone, estimate the pupil's center by using the detected pupil object, and then use the coordinate information to perform and predict the gaze points. Compared with the previous image processing methods of tracking the pupil at the visible-light mode, the proposed method can detect the pupil object at any eyeball position, and it also is not affected by light and shadow interferences in real applications.

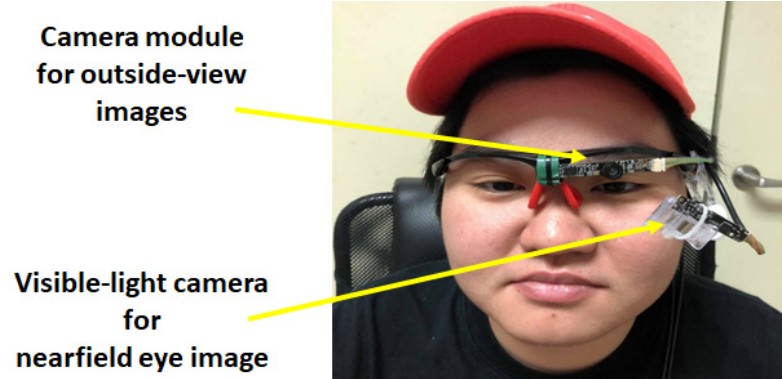

**Figure 4.** The self-made visible-light wearable gaze tracking device.

### 3.1. Pre-Processing for Pupil Center Estimation and Tracking

In the pre-processing work, the visible-light nearfield eye images are recorded for datasets, and the collected eye image datasets will be labeled for training and testing the trained inference model. Moreover, we choose several YOLO-based deep-learning models for pupil object detection. By using the YOLO-based deep-learning model to detect the pupil's object, the proposed system can track and estimate the center coordinate of pupil box for calibration and gaze prediction.

### 3.1.1. Generation of Training and Validation Datasets

In order to collect nearfield eye images from eight different users for training and testing, each user used a self-made wearable eye tracker to emulate the gaze tracking operations, after which the users rotated their gaze around the screen to capture the nearfield eye images with various eyeball positions. The collected and recorded near-eye images cover all possible eyeball positions. Figure 5 illustrates nearfield visible-light eye images collected for training and in-person tests. In Figure 5, the numbers of near-eye images from person 1, person 2, person 3, person 4, person 5, person 6, person 7, and person 8 are 849, 987, 734, 826, 977, 896, 886, and 939, respectively. Moreover, the near-eye images captured from person 9 to person 16 are used for the cross-person test. Thus, the nearfield eye images from person 1 to person 8 without closed eyes are selected, of which approximately 900 images per person are used for training and testing. The image datasets consisted of a total of 7094 eye patterns. Next, these selected eye images will go through a labeling process before training the inference model for pupil object detection and tracking. To increase the number of images in the training phase, the data augmentation process, which randomly changes angle, saturation, exposure, and hue of the selected image dataset, is enabled by using the framework of YOLO. After the 160,000 iterations for training the YOLOv3-tiny-based models, the total number of images used for training will be up to 10 million.

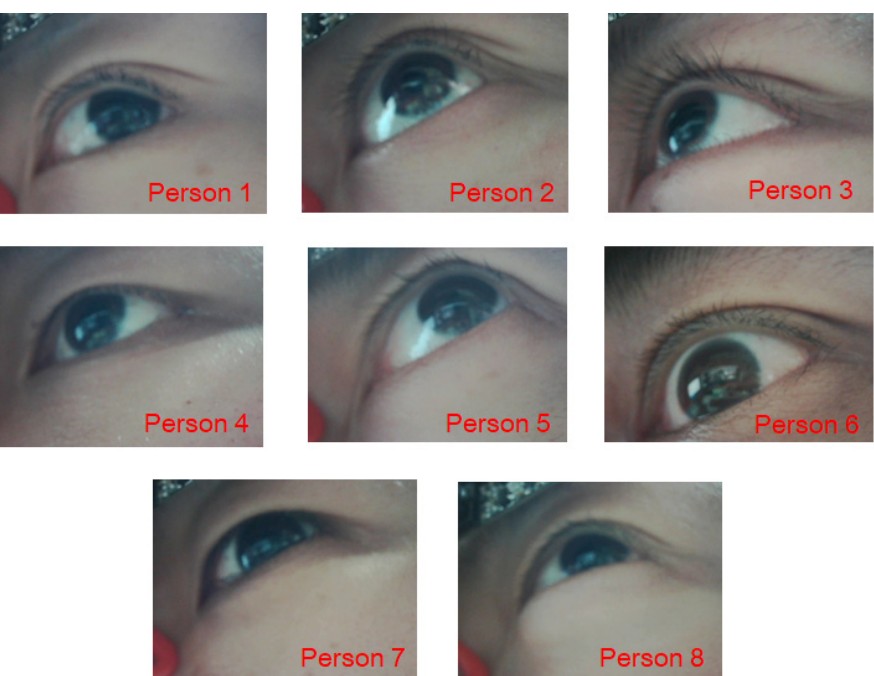

**Figure 5.** Collected datasets of nearfield visible-light eye images.

Before training the model, the designer must find the correct image object (or answer) in the image datasets, to train the model what to learn and to identify the location of the correct object in the whole image, which is commonly known as the ground truth, so the labeling process must be done before the training process. The position of the pupil center varies in the different nearfield eye images. Figure 6 demonstrates the labeling process for the location of the pupil's ROI object. For the following learning process of the detector of the pupil zone, the size of the labeled box also affects the detection accuracy of the network architecture. In our experiments, firstly, a labeled box with the $10 \times 10$ pixels size is tested, and then, a labeled box with a size ranging from $20 \times 20$ pixels to $30 \times 30$ pixels is adopted. The size of the labeled bounding box is closely related to the number of features analyzed by the deep-learning architecture. In the labeling process [40], in order to improve the accuracy of the ground truth, the ellipse fitting method was applied for

the pre-positioning of the pupil's center, after which we used a manual method to further correct the coordinates of the pupil center in each selected nearfield eye image. Before the training process, all labeled images (i.e., 7094 eye images) are randomly divided into a training part, verification part, and test part, with corresponding ratios of 8:1:1. Therefore, the number of labeled eye images used for training is 5676, and the number of labeled images used for verification and testing is 709.

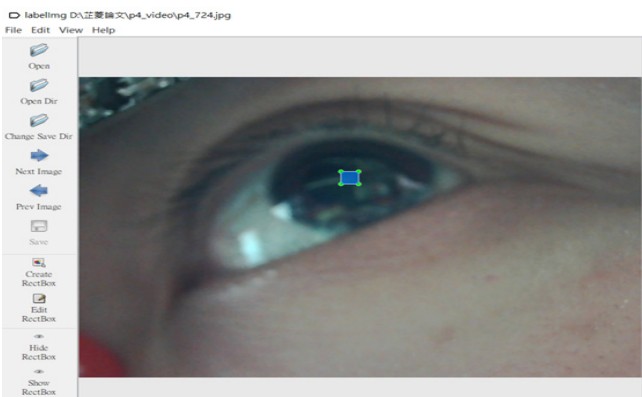

**Figure 6.** The labeling process for location of the pupil's ROI.

### 3.1.2. The Used YOLO-Based Models and Training/Validation Process

The proposed design uses the YOLOv3-tiny deep-learning model. The reasons for choosing a network model based on YOLOv3-tiny are: (1) The YOLOv3-tiny-based model achieves good detection and recognition performance at the same time; (2) The YOLOv3-tiny-based model is effective for small object detection, and the pupil in the nearfield eye image is also a very small object. In addition, because the number of convolutional layers in YOLOv3-tiny is small, the inference model based on YOLOv3-tiny can achieve real-time operations on the software-based embedded platform.

In the original YOLOv3-tiny model, as shown in Figure 7, each layer uses three anchor boxes to predict the bounding boxes in a grid cell. However, only one pupil object category needs to be detected in each nearfield eye image, after which the number of anchor boxes required can be reduced. Figure 8 illustrates the network architecture of the YOLOv3-tiny model with only one anchor. Besides that the number of required anchor boxes can be reduced, since the image features of the detected pupil object is not intricate, the number of convolutional layers can also be reduced. Figure 9 shows the network architecture of the YOLOv3-tiny model with one anchor and a one-way path. In Figure 9, since the number of convolutional layers is reduced, the computational complexity can also be reduced effectively.

Before the labeled eye images are fed into the YOLOv3-tiny-based deep-learning models, the parameters used for training deep-learning networks must be set properly, e.g., batch, decay, subdivision, momentum, etc., after which the inference models for pupil center tracking will be achieved accurately. In our training design for models, the parameters of Width/height, Channels, Batch, Max_batches, Subdivision, Momentum, Decay, and Learning_rate are set to 416, 3, 64, 500200, 2, 0.9, 0.0005, and 0.001, respectively.

In the architecture of the YOLOv3-tiny model, 13 convolution layers with different filter numbers and pooling layers with different strides are established to allow the YOLOv3-tiny model to have a high enough learning capability for training from images. In addition, the YOLOv3-tiny-based model not only uses the convolution neural network, but also adds upsampling and concrete calculation methods. During the learning process, the YOLO model uses training datasets for training, uses the images in the validation dataset to test the model, observes whether the model is indeed learning correctly, uses the weight back propagation method to gradually converge the weight, and observes whether the weight converges with the loss function.

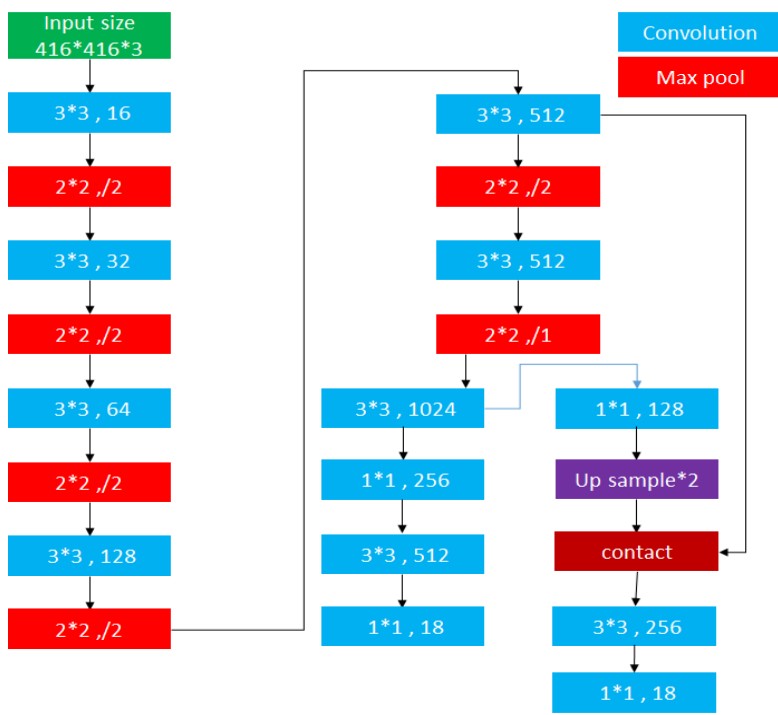

**Figure 7.** The network architecture of the original YOLOv3-tiny model.

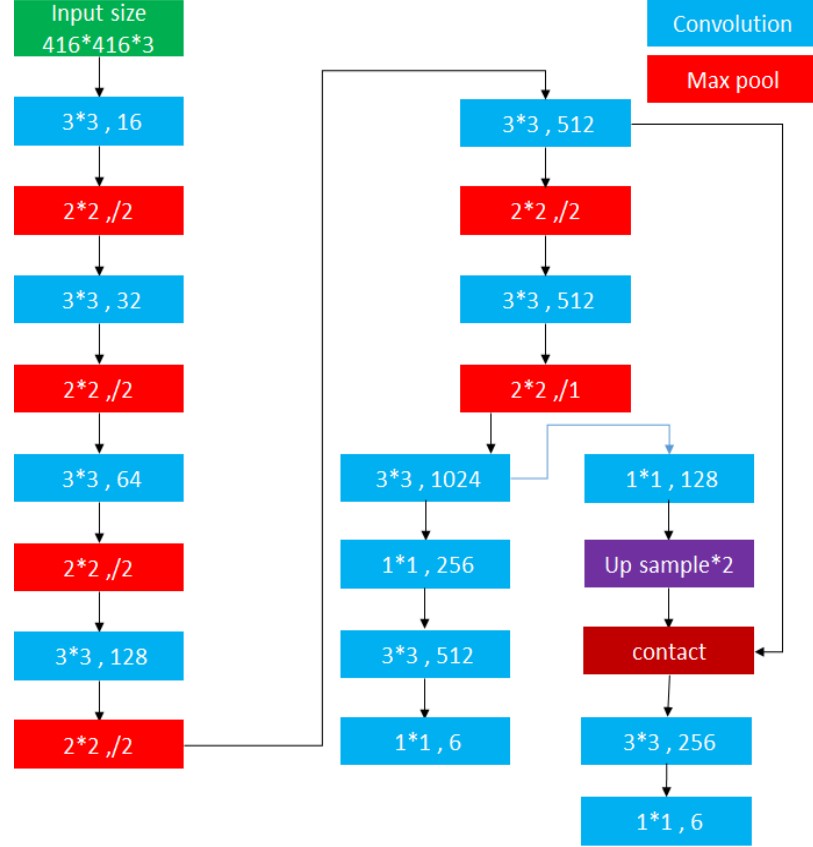

**Figure 8.** The network architecture of the YOLOv3-tiny model with one anchor.

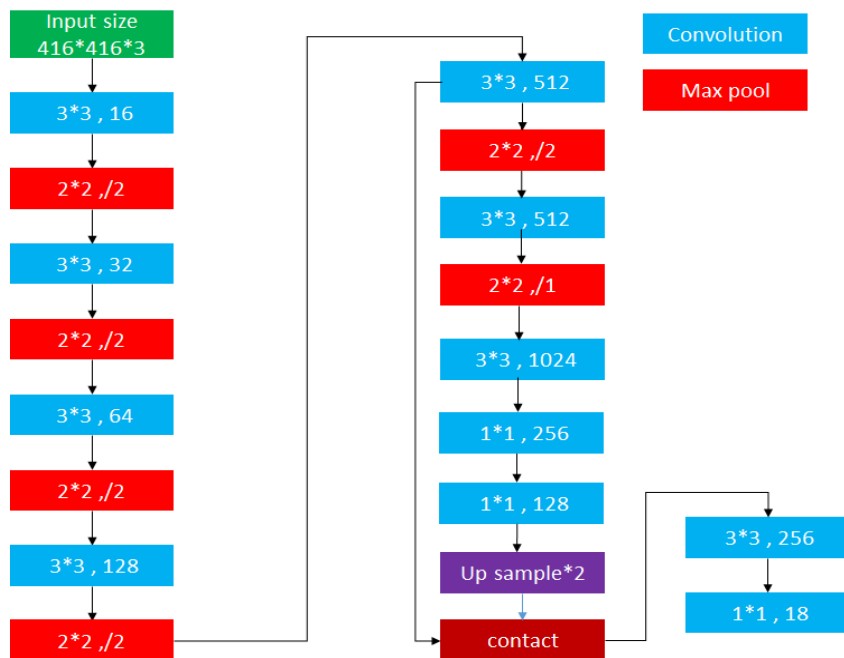

**Figure 9.** The network architecture of the YOLOv3-tiny model with one anchor and a one-way path.

Figure 10 depicts the training and testing processes by the YOLO-based models for tracking pupils. In the detection and classification process by the YOLO model, the grid cell, anchor box, and activation function are used to determine the position of the object bounding box in the image. Therefore, the well-trained model can detect the position of the pupil in the image with the bounding box. The detected pupil box can be compared with the ground truth to generate the intersection over union (IoU), precision, and recall values; we use these values to judge the quality of the model training.

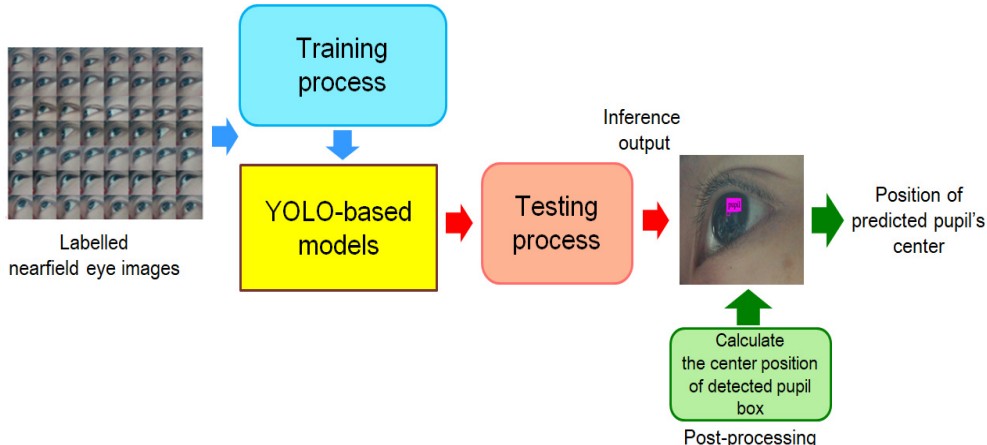

**Figure 10.** The training and testing processes by the YOLO-based models for tracking pupil.

Table 2 presents the confusion matrix used for the performance evaluation of classification. In pattern recognition for information retrieval and classification, recall is mentioned to as the true positive rate or sensitivity, and precision is mentioned as the positive predictive value.

**Table 2.** Confusion matrix for classification.

| | | Actual Values | |
|---|---|---|---|
| | | **Positives** | **Negatives** |
| Predictive values | Positives | True Positives (TP) | False Positives (FP) |
| | Negatives | False Negatives (FN) | True Negatives (TN) |

Both precision and recall are based on an understanding and measurement of relevance. Based on Table 1, the definition of precision is expressed as follows:

$$\text{Precision} = TP/(TP + FP), \tag{1}$$

and recall is defined as follows:

$$\text{Recall} = TP/(TP + FN). \tag{2}$$

After using the YOLO-based deep-learning model to detect the pupil's ROI object, the midpoint coordinate of the pupil's bounding box can be found, and the midpoint coordinate value is the estimated pupil's center point. The midpoint coordinates for the estimated pupil's centers are evaluated with Euclidean distance by comparing the estimated coordinates with the coordinates of ground truth, and the tracking errors of pupil's center will also be obtained for the performance evaluation. Using the pupil's central coordinates for the coordinate conversion of calibration, the proposed wearable device can correctly estimate and predict the outward gaze points.

### 3.2. Calibration and Gaze Tracking Processes

Before using the wearable eye tracker to predict the gaze points, the gaze tracking device must undergo a calibration procedure. Because the appearance of each user's eye and the application distance to the near-eye camera are different, the calibration must be done to confirm the coordinate mapping scheme between the pupil's center and the outward scene, after which the wearable gaze tracker can correctly predict the gaze points correspond to the outward scene. Figure 11 depicts the used calibration process of the proposed wearable gaze tracker. The steps of the applied calibration process are described as follows:

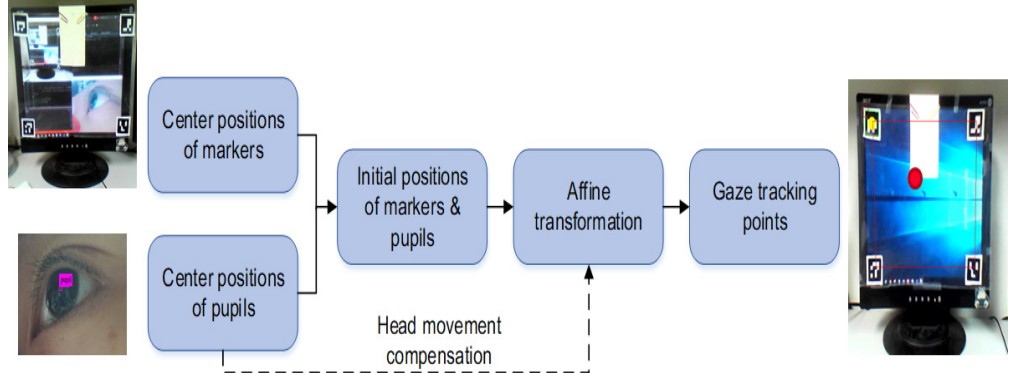

**Figure 11.** The used calibration process of the proposed wearable gaze tracker.

At the first step, the system collects the correspondence between the center point coordinates of the pupils captured by the nearfield eye camera and the gaze points simulated by the outward camera. Before calibration, in addition to recording the positions of the center points of the pupil movement, the trajectory of markers captured by the outward camera must also be recorded. Figure 12 shows the trajectory diagram of tracking the pupil's center with the nearfield eye image for calibration. In addition, Figure 12 reveals the

corresponding trajectory diagram of tracking four markers' center points with the outward image for calibration.

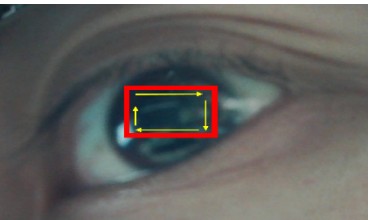

**Figure 12.** Diagram of tracking pupil's center with the nearfield eye camera for calibration.

In Figure 12, the yellow line indicates the sequence of the movement path of gazing the markers, and the red rectangle indicates the range of the eye movement. In Figure 13, when the calibration process is executed, the user gazes at the four markers sequentially from ID 1, ID 2, ID 3, to ID 4, and the system records the coordinates of the pupil movement path while watching the gazing coordinates of the outward scene with markers. Then, the proposed system performs the coordinate conversion to achieve the calibration effect.

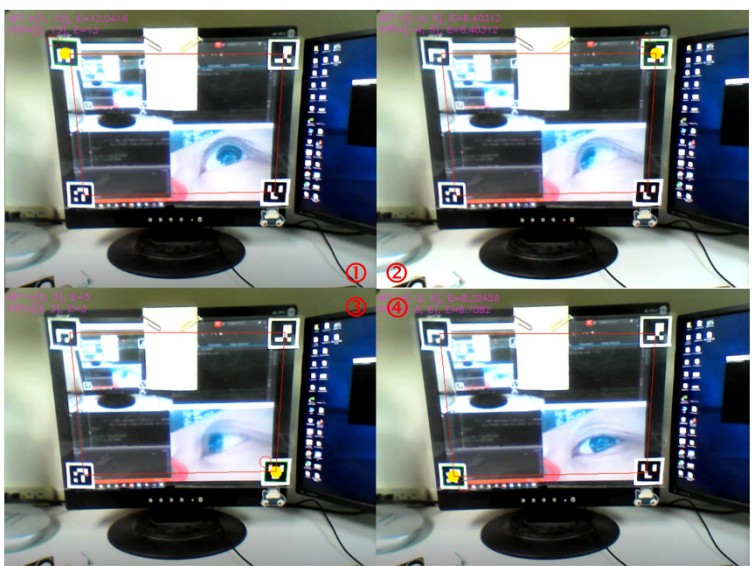

**Figure 13.** Diagram of tracking four markers' center points with the outward camera for calibration ① When the user gazes at the marker ID1 for calibration. ② When the user gazes at the marker ID2 for calibration. ③ When the user gazes at the marker ID3 for calibration. ④ When the user gazes at the marker ID4 for calibration.

The pseudo-affine transform is used for the coordinate conversion between the pupil and the outward scene. The equation of the pseudo-affine conversion is expressed as follows:

$$
\begin{bmatrix} X \\ Y \end{bmatrix} = \begin{bmatrix} a_1 & a_2 & a_3 & a_4 \\ a_5 & a_6 & a_7 & a_8 \end{bmatrix} \begin{bmatrix} X_p \\ Y_p \\ X_pY_p \\ 1 \end{bmatrix},
\tag{3}
$$

where $(X, Y)$ is the outward coordinate of the markers' center point coordinate system, and $(X_p, Y_p)$ is the nearfield eye coordinates of the pupils' center point coordinate system. In (3), the unknown transformation coefficients of $a_1 \sim a_8$ need to be solved by the coordinate values computed and estimated from $(X, Y)$ and $(X_p, Y_p)$. In order to carry out the calculation of the transformation parameters of $a_1 \sim a_8$ for calibration, Equation (4) describes the pseudo-affine transformation with four-points calibration, which is used to solve the transformation

coefficients of the coordinate conversion. The equation of the pseudo-affine conversion with four-point calibration is expressed as follows:

$$
\begin{bmatrix}
X_1 \\
Y_1 \\
X_2 \\
Y_2 \\
X_3 \\
Y_3 \\
X_4 \\
Y_4
\end{bmatrix}
=
\begin{bmatrix}
X_{p1} & Y_{p1} & X_{p1}Y_{p1} & 1 & 0 & 0 & 0 & 0 \\
0 & 0 & 0 & 0 & X_{p1} & Y_{p1} & X_{p1}Y_{p1} & 1 \\
X_{p2} & Y_{p2} & X_{p2}Y_{p2} & 1 & 0 & 0 & 0 & 0 \\
0 & 0 & 0 & 0 & X_{p2} & Y_{p2} & X_{p2}Y_{p2} & 1 \\
X_{p3} & Y_{p3} & X_{p3}Y_{p3} & 1 & 0 & 0 & 0 & 0 \\
0 & 0 & 0 & 0 & X_{p4} & Y_{p4} & X_{p4}Y_{p4} & 1 \\
X_{p4} & Y_{p4} & X_{p4}Y_{p4} & 1 & 0 & 0 & 0 & 0 \\
0 & 0 & 0 & 0 & X_{p4} & Y_{p4} & X_{p4}Y_{p4} & 1
\end{bmatrix}
\cdot
\begin{bmatrix}
a_1 \\
a_2 \\
a_3 \\
a_4 \\
a_5 \\
a_6 \\
a_7 \\
a_8
\end{bmatrix},
\tag{4}
$$

where $(X_1, Y_1)$, $(X_2, Y_2)$, $(X_3, Y_3)$, and $(X_4, Y_4)$ are the outward coordinates of the four marker's center points and $(X_{p1}, Y_{p1})$, $(X_{p2}, y_{p2})$, $(X_{p3}, y_{p3})$, and $(X_{p4}, y_{p4})$ are the nearfield eye coordinates of the four pupil's center points, respectively.

## 4. Experimental Results and Comparisons

In the experimental environment, the general illuminance values measured by the spectrometer are 203.68lx. To verify the proposed pupil tracking design under a strong light environment, we use a desk lamp to create the effect, and the measurement values of illuminances are 480.22lx. On the contrary, for verifying the proposed pupil tracking design under a darker environment, the illuminance values of the experimental environment are set to 86.36lx. In addition to achieving real-time operation effects, the used YOLOv3-tiny-based deep-learning models can solve the problem of light and shadow interferences that cannot be solved effectively by the traditional pupil tracking methods. In our experiments, a computing server with two NVIDIA GPU accelerator cards was used to train the pupil tracking models, and a personal computer with a 3.4 GHz operating frequency CPU and an NVIDIA GPU card was used to test the YOLO-based inference models. The visible-light wearable gaze tracker utilized two camera modules for shooting, one of which is an outward scene camera module with a resolution of $1600 \times 1200$ pixels, and the other is a visible-light nearfield eye camera module, which provides a resolution of $1280 \times 720$ pixels and can reach 30 frames per second.

Through the proposed YOLOv3-tiny-based deep-learning methods to track the position of the pupil center, the size of the pupil's detection box is between $20 \times 20$ and $30 \times 30$ pixels; the pupil center at any position of the eyeball can be correctly inferred, and the inference results are shown in Figure 14. Table 3 lists the performance comparison of visible-light pupil object detection among the three YOLO-based models. In Table 3, the three YOLO-based deep-learning-based models perform the pupil detection precision, which is up to 80%, and the recall rate is up to 83%.

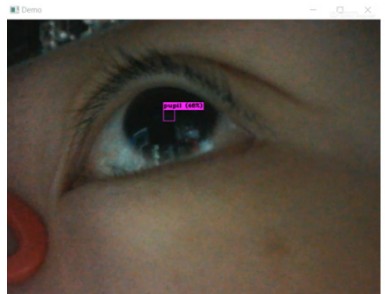 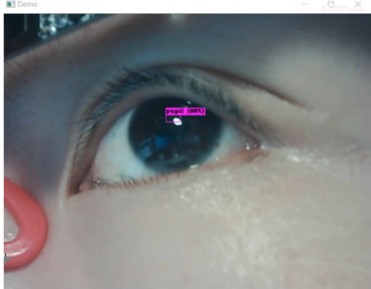

**Figure 14.** The inference results by using the YOLOv3-tiny-based model.

**Table 3.** Performance comparisons of visible-light pupil object detection among the three YOLO-based models.

| Model | Validation | | Test | |
|---|---|---|---|---|
| | Precision | Recall | Precision | Recall |
| YOLOv3-tiny (original) | 81.40% | 83% | 78.43% | 81% |
| YOLOv3-tiny (one anchor) | 80.19% | 84% | 80.50% | 83% |
| YOLOv3-tiny (one anchor/one way) | 79.79% | 82% | 76.82% | 81% |

For the tested eye images, to evaluate the accuracy of the trained YOLOv3-tiny-based models, Table 4 lists the comparison results of visible-light pupils' center tracking errors between the YOLOv3-tiny and the previous designs without using deep-learning technologies. In Table 4, the applied YOLOv3-tiny-based design performs smaller pupil tracking errors than the previous designs in [15,21]. The average pupil tracking error of the proposed deep-learning-based model is less than 2 pixels, which is much smaller than the pupil tracking error of the previous ellipse fitting design in [15]. Compared with the previous design in [15,21], without using deep-learning technology, the proposed deep-learning-based design has very small pupil tracking errors under various visible-light conditions.

**Table 4.** Comparisons of the visible-light pupils' center tracking errors between the YOLOv3-tiny and the previous ellipse fitting based designs.

| Methods | Ellipse Fitting Based Design [15] | Gradients-Based Design [21] | YOLOv3-Tiny (Original) |
|---|---|---|---|
| Average tracking errors (pixels) | Larger than 50 | 5.99 | 1.62 |
| Standard deviation (pixels) | 63.1 | N/A | 2.03 |
| Max/min errors (pixels) | 294/0 | 40/0 | 21/0 |

To verify the pupil tracking errors of the different YOLOv3-tiny-based deep-learning models, Table 5 lists the comparison results of the visible-light pupils' center tracking errors among the three YOLO-based models, where the datasets of person 1 to person 8 are used for the training mode. In Table 5, the average pupil tracking errors of the three YOLOv3-tiny-based deep-learning-based models are less than 2 pixels. Table 6 lists the comparison results of the visible-light pupils' center tracking errors among the three YOLO-based models for the cross-person testing mode. In Table 6, the average pupil tracking errors of the three YOLOv3-tiny-based deep-learning-based models are less than 5 pixels.

**Table 5.** Comparisons of the visible-light pupils' center tracking errors among the three YOLO-based models by using the datasets of person 1 to person 8 for the training mode.

| Model | Mean Errors (Pixels) | Variance (Pixels) |
|---|---|---|
| YOLOv3-tiny (original) | 1.62 | 2.03 |
| YOLOv3-tiny (one anchor) | 1.58 | 1.87 |
| YOLOv3-tiny (one anchor/one way) | 1.46 | 1.75 |

**Table 6.** Comparisons of the visible-light pupils' center tracking errors among the three YOLO-based models by using the datasets of person 9 to person 16 for the cross-person testing mode.

| Model | Mean Errors (Pixels) | Variance (Pixels) |
|---|---|---|
| YOLOv3-tiny (original) | 4.11 | 2.31 |
| YOLOv3-tiny (one anchor) | 3.99 | 2.27 |
| YOLOv3-tiny (one anchor/one way) | 4.93 | 2.31 |

Table 7 describes the comparison of computational complexities among the three YOLOv3-tiny-based models. In Table 7, "BFLOPS" means billion float operations per second. Using a personal computer with an Nvidia GeForce RTX 2060 card, the processing frame rate of pupil tracking with YOLOv3-tiny-based models is up to 60 FPS. Using a personal computer with an Nvidia GeForce RTX 2080 card, the processing frame rate of pupil tracking with YOLOv3-tiny-based models is up to 150 FPS. Moreover, when using the NVIDIA XAVIER embedded platform [41], the processing frame rate of pupil tracking with YOLOv3-tiny-based models is up to 88 FPS.

**Table 7.** Comparisons of computational complexities among the three YOLO-based models.

| Model | BFLOPS |
|---|---|
| YOLOv3-tiny (original) | 5.448 |
| YOLOv3-tiny (one anchor) | 5.441 |
| YOLOv3-tiny (one anchor/one way) | 5.042 |

Figure 15 illustrates the calculation of the gaze tracking errors, and the expression of the calculation method is described in Equation (5) as follows:

$$\theta = \tan^{-1} \frac{E}{D} \tag{5}$$

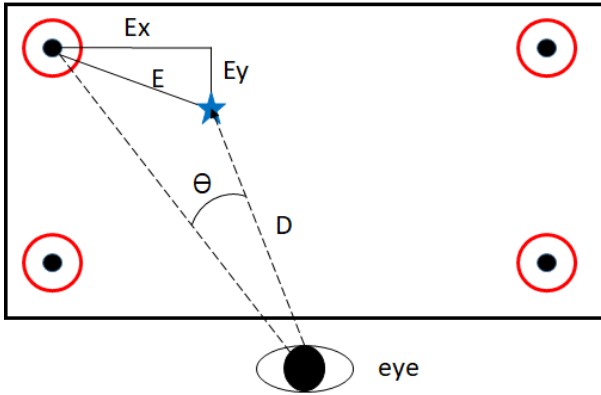

**Figure 15.** Diagram of estimating gaze tracking errors.

In Equation (5), the angle $\theta$ reflects the gaze tracking errors (i.e., degrees), E is the absolute error (by pixels) at the gaze plane, and D is the application distance between the user and the gaze plane, where D is set to 55 cm. For the evaluation of the gaze tracking errors, Figure 16a depicts four target markers used for calibration for the training mode, and Figure 16b presents five target points used for estimation for the testing mode. For the training mode, using the four target markers shown in Figure 16a, Table 8 lists the

comparison of gaze tracking errors among the three applied YOLOv3-tiny-based models. After being joined with the calibration process, the average gaze tracking errors by the proposed YOLOv3-tiny-based pupil tracking models are less than 2.9 degrees for the training mode. Table 9 lists the comparison of the gaze tracking errors among the three YOLOv3-tiny-based models for the testing mode using the five target points shown in Figure 16b. In Table 9, the average gaze tracking errors by the proposed YOLOv3-tiny-based pupil tracking models are less than 3.5 degrees.

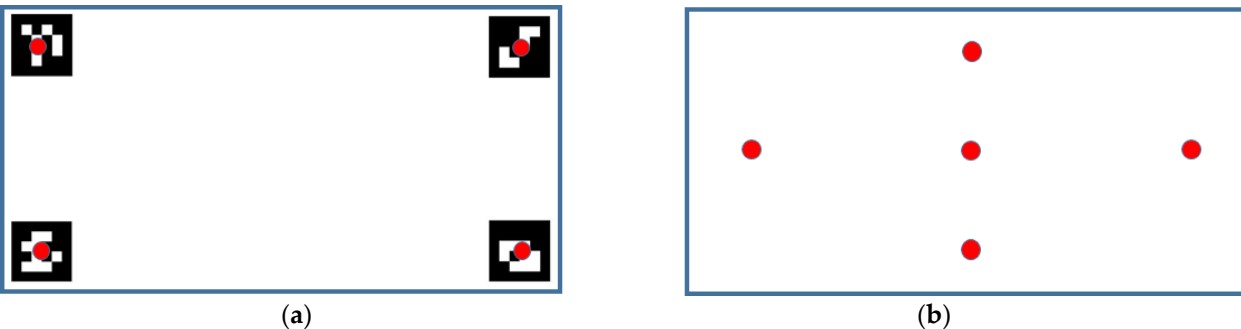

(**a**)　　　　　　　　　　　　　　　　　　　　　　　　(**b**)

**Figure 16.** (**a**) Four target markers used for calibration for the training mode. (**b**) Five target points used for estimation for the testing mode.

**Table 8.** Comparisons of gaze tracking errors among the three YOLOv3-tiny-based models for the training mode.

| Model | Mean Gaze Tracking Errors, θ (Degrees) | Variance (Degrees) |
|---|---|---|
| YOLOv3-tiny (original) | 2.131 | 0.249 |
| YOLOv3-tiny (one anchor) | 2.22 | 0.227 |
| YOLOv3-tiny (one anchor/one way) | 2.959 | 0.384 |

**Table 9.** Comparisons of gaze tracking errors among the three YOLOv3-tiny-based models for the testing mode.

| Model | Mean Gaze trAcking Errors, θ (Degrees) | Variance (Degrees) |
|---|---|---|
| YOLOv3-tiny (original) | 2.434 | 0.468 |
| YOLOv3-tiny (one anchor) | 2.57 | 0.268 |
| YOLOv3-tiny (one anchor/one way) | 3.523 | 0.359 |

Table 10 revealed the performance comparison among the different deep-learning-based gaze tracking designs. In general, the gaze tracking errors by the wearable eye tracker will be less than those by the non-wearable eye trackers. For the wearable eye tracker designs, since the infrared-like oculography images have a higher contrast and less light, noise, and shadow interferences than the near-eye visible-light images, the method in [28] performs smaller pupil tracking and gaze estimation errors in comparison with our proposed design. However, when the comparison is focused on the setup with near-eye visible-light images, the proposed YOLOv3-tiny-based pupil tracking methods can provide the best, state-of-the-art pupil tracking accuracy.

**Table 10.** Performance comparison of the deep-learning-based gaze tracking designs.

| Deep-Learning-Based Methods | Operational Mode/Setup | Pupil Tracking Errors | Gaze Estimation Errors |
|---|---|---|---|
| Sun et al. [22] | Visible-light mode /Non-Wearable | N/A | Mean errors are less than 7.75 degrees |
| Lemley et al. [24] | Visible-light mode /Non-Wearable | N/A | Less than 3.64 degrees |
| Cha et al. [26] | Visible-light mode /Non-Wearable | N/A | Accuracy 92.4% by 9 gaze zones |
| Zhang et al. [29] | Visible-light mode /Non-Wearable | N/A | Mean errors are 10.8 degrees for cross-dataset /Less than 5.5 degrees for cross-person |
| Lian et al. [32] | Visible-light mode /Non-Wearable | N/A | Less than 5 degrees |
| Li et al. [34] | Visible-light mode /Non-Wearable | N/A | Less than 2 degrees at training mode /Less than 5 degrees at testing mode |
| Rakhmatulin et al. [35] | Visible-light mode /Non-Wearable | N/A | Less than 2 degrees |
| Brousseau et al. [36] | Infrared mode /Non-Wearable | Less than 6 pixels | Gaze-estimation bias is 0.72 degrees |
| Yiu et al. [28] | Near-eye infrared mode/Wearable | Median accuracy is 1.0 pixel | Less than 0.5 degrees |
| Proposed design | Near-eye visible-light mode/Wearable | Less than 5 pixels for the cross-person testing mode | Less than 2.9 degrees for the training mode /Less than 3.5 degrees for the testing mode |

Figure 17 demonstrates the implementation result of the proposed gaze tracking design on the Nvidia Xavier embedded platform [41]. Using the embedded software implementation with Xavier, the proposed YOLOv3-tiny-based visible-light wearable gaze tracking design operates up to 20 frames per second to be suitable for practical consumer applications.

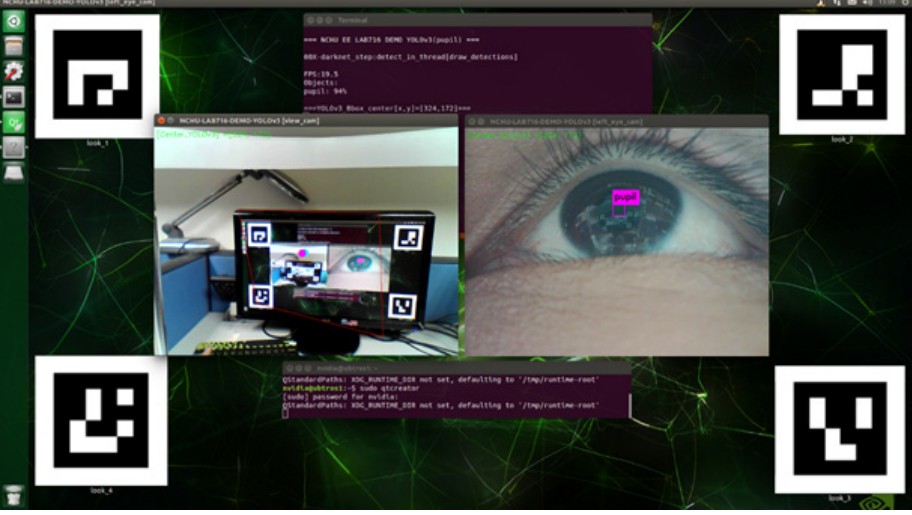

**Figure 17.** Implementation of the proposed gaze tracking design on Nvidia Xavier embedded platform.

## 5. Conclusions and Future Work

For the application of visible-light wearable gaze trackers, YOLOv3-tiny-based deep-learning pupil tracking methodology is developed in this paper. By applying YOLO-based object detection technology, the proposed pupil tracking method effectively estimates and predicts the center of the pupil in a visible-light mode. By using the developed YOLO-based model to test the pupil tracking performance, the accuracy is up to 80%, and the recall rate is close to 83%. In addition, the average visible-light pupil tracking errors of the proposed YOLO based deep-learning design are smaller than 2 pixels for the training mode and 5 pixels for the cross-person testing mode, which are much smaller than those of the previous designs in [15,21], without using deep-learning technology under the same visible-light conditions. After the calibration process, the average gaze tracking errors using the proposed YOLOv3-tiny-based pupil tracking models are smaller than 2.9 and 3.5 degrees for the training and testing modes, respectively. The proposed visible-light wearable gaze tracking design performs up to 20 frames per second on the Nvidia Xavier embedded platform.

In this design, the application distance between the user's head and the screen is fixed during the operation, and the direction of the head pose tries to keep fixed. In future works, a head movement compensation function will be added, and the proposed wearable gaze tracker will be more convenient and friendly for practical uses. To raise the high-precision recognition ability of the pupil location and tracking, the deep-learning model will be updated with online training using the big image databases in the cloud to fit the pupil position for different eye colors and eye textures.

**Author Contributions:** Conceptualization, W.-L.O. and T.-L.K.; methodology, T.-L.K. and C.-C.C.; software and validation, W.-L.O. and T.-L.K.; writing—original draft preparation, T.-L.K. and C.-P.F.; writing—review and editing, C.-P.F.; supervision, C.-P.F. All authors have read and agreed to the published version of the manuscript.

**Funding:** This work was supported by the Ministry of Science and Technology, Taiwan, R.O.C. under Grant MOST 109-2218-E-005-008.

**Informed Consent Statement:** Not applicable.

**Data Availability Statement:** The data presented in this study are available on request from the corresponding author.

**Conflicts of Interest:** The authors declare no conflict of interest.

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
