# Peer review of "Deep-Learning-Based Pupil Center Detection and Tracking Technology for Visible-Light Wearable Gaze Tracking Devices"

_applsci, doi:10.3390/app11020851_

Round 1

Reviewer 1 Report

The paper “Deep Learning Based Pupil Center Detection and Tracking Technology for Visible-Light Wearable Gaze Tracking Devices” describes the application of YOLOv3-tiny model for detection of pupil in the visible light.

I am missing deeper literature review in the paper. I did not found information about similar attempts, how succesfull they were etc.

The methods section is well written, and what has been done is clearly described. I was surprised that only 8 users were used for the training. Authors wrote at the end of the paper, that they plan to test the process for different eye colours and eye textures in the future, but I am curious what will happen if the wearable eye-tracker will be used by someone else than those eight users that were used for training? Will it work somehow?

Caption of figure 15 - typo

Author Response

Response to Reviewer 1 Comments

The paper “Deep Learning Based Pupil Center Detection and Tracking Technology for Visible-Light Wearable Gaze Tracking Devices” describes the application of YOLOv3-tiny model for detection of pupil in the visible light.  

Point 1: I am missing deeper literature review in the paper. I did not found information about similar attempts, how successful they were etc.  

Response 1: Thank you very much for the review’s comments.

We added the information about similar attempts, and please refer the updated and added contents, which are located from Line 62 to Line 165 in the revision.  

Besides, in the revision, Table 1 shows the overview of the deep-learning based gaze tracking designs, and Table 10 also reveals the performance comparison among the different deep-learning based gaze tracking designs.

Point 2: The methods section is well written, and what has been done is clearly described.

Response 2: Thank you very much for the review’s positive comments.

Point 3:  I was surprised that only 8 users were used for the training. Authors wrote at the end of the paper, that they plan to test the process for different eye colours and eye textures in the future, but I am curious what will happen if the wearable eye tracker will be used by someone else than those eight users that were used for training? Will it work somehow?

Response 3: Thank you very much for the review’s comments.

In the revision, the authors add Table 6.  Table 6 lists the comparisons of visible-light pupil’s center tracking errors among the three YOLO-based models by using the datasets of person 9 to person 16 at the cross-person testing mode.

Point 4: Caption of figure 15 - typo

Response 4: Thank you very much for the review’s comments.

The authors correct it, and please refer the content, which is located at Line 472 in the revision.

Reviewer 2 Report

The paper proposes a deep learning-based pupil center detecting and gaze tracking system. A convolution neural network is used to detect the pupil from RGB images captured from a wearable device. YOLO network has been utilized through transfer learning in this regard. The experimental results validate that the proposed method can outperform the existing pupil tracking system that uses non-deep learning methods.

It is obvious to expect that a deep learning method has a higher recognition accuracy than a non-deep learning method. Furthermore, many developments of deep learning-based pupil tracking methods could be seen in the literature (e.g., [19]-[33]). Thus, I feel the experimental comparison must be performed, considering deep learning-based pupil tracking methods seen in the literature. Apart from that, the novelty is not clear with respect to the state of the art, and it should be clearly outlined.

The data set used for the training and testing was collected with only eight persons. I would suggest authors collect data from more persons. Was the tracking accuracy tested considering the same persons? I would recommend the authors consider some other persons to assess the accuracy in that case and present the results in the manuscript.

In the comparisons of results, I would suggest the authors include statistical conclusions. For example, use an ANOVA test to evaluate the significance.

Overall, I feel that the paper needs a major revision before considering for the publication.

Author Response

Response to Reviewer 2 Comments

The paper proposes a deep learning-based pupil center detecting and gaze tracking system. A convolution neural network is used to detect the pupil from RGB images captured from a wearable device. YOLO network has been utilized through transfer learning in this regard. The experimental results validate that the proposed method can outperform the existing pupil tracking system that uses non-deep learning methods.

Point 1:  It is obvious to expect that a deep learning method has a higher recognition accuracy than a non-deep learning method. Furthermore, many developments of deep learning-based pupil tracking methods could be seen in the literature (e.g., [19]-[33]). Thus, I feel the experimental comparison must be performed, considering deep learning-based pupil tracking methods seen in the literature.

Response 1:  Thank you very much for the review’s comments.

The authors add the information and descriptions for similar literatures, and please refer the updated and added contents, which are located from Line 62 to Line 165 in the revision.  

Besides, in the revision, Table 1 shows the overview of the deep-learning based gaze tracking designs, and Table 10 also reveals the performance comparison among the different deep-learning based gaze tracking designs.

Point 2: Apart from that, the novelty is not clear with respect to the state of the art, and it should be clearly outlined.

Response 2:  Thank you very much for the review’s comments.

The authors note the novelty and advantages of the proposed design in the revision, and please refer the updated contents, which are located from Line 174 to Line 189 in the revision.

The main contribution of this study is that the pupil location in nearfield visible-light eye images is detected precisely by the YOLOv3-tiny based deep learning technology.  By our best knowledge from surveying in related studies, the novelty of this work is to provide the leading reference design of deep-learning based pupil tracking technology using the nearfield visible-light eye images for the application of wearable gaze tracking devices. 

Besides, the other novelties and advantages of the proposed deep-learning based pupil tracking technology are described as follows:

(1)By using a nearfield visible-light eye image dataset for training, the used deep learning models establish to achieve real-time and accurate detection of the pupil position at the visible-light mode.

(2)The proposed design detects the position of pupil’s object at any eyeball movement condition, which is more effective than the traditional image processing methods without deep learning at the near-eye visible-light mode.

(3)The proposed pupil tracking technology can overcome efficiently the light and shadow interferences at the near-eye visible-light mode, and the detection accuracy of pupil’s location is higher than that of previous wearable gaze tracking designs without using deep-learning technologies.

Point 3:  The data set used for the training and testing was collected with only eight persons. I would suggest authors collect data from more persons.

Response 3:  Thank you very much for review’s comments.

Although the total number of nearfield eye images is 7,094 eye patterns are selected from person 1 to person 8, to increase the number of images in the training phase, the data augmentation process, which randomly changes angle, saturation, exposure, and hue of the selected image dataset, is enabled by using the framework of YOLO.  After the 160,000 iterations for training the YOLOv3-tiny based models, the total number of images used for training will be up to 10 millions. 

Besides, the near-eye images captured from person 9 to person 16 are added for the cross-person test.

Point 4: Was the tracking accuracy tested considering the same persons? I would recommend the authors consider some other persons to assess the accuracy in that case and present the results in the manuscript.

Response 4: Thank you very much for review’s comments.

In the revision, the authors add Table 6.  Table 6 lists the comparisons of visible-light pupil’s center tracking errors among the three YOLO-based models by using the datasets of person 9 to person 16 at the cross-person testing mode.

Point 5: In the comparisons of results, I would suggest the authors include statistical conclusions. For example, use an ANOVA test to evaluate the significance.

Response 5: Thank you very much for review’s comments.

The authors add the analysis of variance (ANOVA) in the revision. Please refer Tables 4, 5, 6, 8, and 9 in the revision.

Point 6: Overall, I feel that the paper needs a major revision before considering for the publication.

Response 6: Thank you very much for the review’s comments.

By following all of the reviewers’ comments, we have done the major revision.

Round 2

Reviewer 1 Report

I would like to thank the authors for they work on the manuscript. I like especially the way how they extend the introduction. I am now satisfied with the paper.  

Author Response

Response to Reviewer 1 Comments

I would like to thank the authors for they work on the manuscript. I like especially the way how they extend the introduction. I am now satisfied with the paper.    

Response: The authors very appreciate for the review’s comments to improve the quality of this paper.

According to Academic Editor’s notes, in the revision, the authors have added and highlighted the three papers, which are listed in [4]-[6] as follows:

[4] Katona, J.;Kovari, A.;Costescu, C.;Rosan, A.;Hathazi, A.;Heldal, I.;Helgesen, C.;Thill, S. The Examination Task of Source-code Debugging Using GP3 Eye Tracker. 10th International Conference on Cognitive Infocommunications (CogInfoCom), 2019;pp. 329-334.

[5] Katona, J.;Kovari, A.;Heldal, I.;Costescu, C.;Rosan, A.;Demeter, R.;Thill, S.;Stefanut, T.  Using Eye-Tracking to Examine Query Syntax and Method Syntax Comprehension in LINQ. 11th International Conference on Cognitive Infocommunications (CogInfoCom), 2020;pp. 437-444.

[6] KĹ‘vári, A.;Katona, J.;  Costescu, C. Evaluation of Eye-Movement Metrics in a Software Debugging Task using GP3 Eye Tracker. Acta Polytechnica Hungarica. 2020, 17, 2, 57-76.

The three papers are added to highlight the application of the eye tracker for the important cognitive information and processes in Section 1.

In the revision, the updated parts are marked by the green color.

Reviewer 2 Report

The effort put by the authors in revising the manuscript is commendable. Most of my concerns are adequately addressed in the revised version. I recommend to accept the paper.

Author Response

Response to Reviewer 2 Comments

The effort put by the authors in revising the manuscript is commendable. Most of my concerns are adequately addressed in the revised version. I recommend to accept the paper.

Response: The authors very appreciate for the review’s comments to improve the quality of this paper.

According to Academic Editor’s notes, in the revision, the authors have added and highlighted the three papers, which are listed in [4]-[6] as follows:

[4] Katona, J.;Kovari, A.;Costescu, C.;Rosan, A.;Hathazi, A.;Heldal, I.;Helgesen, C.;Thill, S. The Examination Task of Source-code Debugging Using GP3 Eye Tracker. 10th International Conference on Cognitive Infocommunications (CogInfoCom), 2019;pp. 329-334.

[5] Katona, J.;Kovari, A.;Heldal, I.;Costescu, C.;Rosan, A.;Demeter, R.;Thill, S.;Stefanut, T.  Using Eye-Tracking to Examine Query Syntax and Method Syntax Comprehension in LINQ. 11th International Conference on Cognitive Infocommunications (CogInfoCom), 2020;pp. 437-444.

[6] KĹ‘vári, A.;Katona, J.;  Costescu, C. Evaluation of Eye-Movement Metrics in a Software Debugging Task using GP3 Eye Tracker. Acta Polytechnica Hungarica. 2020, 17, 2, 57-76.

The three papers are added to highlight the application of the eye tracker for the important cognitive information and processes in Section 1.

In the revision, the updated parts are marked by the green color.